# The Application of Surface Electromyography Technology in Evaluating Paraspinal Muscle Function

**DOI:** 10.3390/diagnostics14111086

**Published:** 2024-05-24

**Authors:** Moran Suo, Lina Zhou, Jinzuo Wang, Huagui Huang, Jing Zhang, Tianze Sun, Xin Liu, Xin Chen, Chunli Song, Zhonghai Li

**Affiliations:** 1Department of Orthopedics, First Affiliated Hospital of Dalian Medical University, Dalian 116011, China; ssuomoran@163.com (M.S.); wangjinzuo0311@163.com (J.W.); huanghuaguispine@163.com (H.H.); zhangjingspine@163.com (J.Z.); suntianze1997@163.com (T.S.); drliuxin9606@163.com (X.L.); 2Key Laboratory of Molecular Mechanism for Repair and Remodeling of Orthopedic Diseases, Dalian 116000, China; 3Department of Neurology, First Affiliated Hospital of Dalian Medical University, Dalian 116011, China; 18098876296@163.com; 4Musculoskeletal Research Laboratory, Department of Orthopaedics & Traumatology, Faculty of Medicine, The Chinese University of Hong Kong, Hong Kong, China; benjaminchen@link.cuhk.edu.hk

**Keywords:** paraspinal muscles, surface electromyography, low back pain, spinal disorders, functional exercise, muscle function

## Abstract

Surface electromyography (sEMG) has emerged as a valuable tool for assessing muscle activity in various clinical and research settings. This review focuses on the application of sEMG specifically in the context of paraspinal muscles. The paraspinal muscles play a critical role in providing stability and facilitating movement of the spine. Dysfunctions or alterations in paraspinal muscle activity can lead to various musculoskeletal disorders and spinal pathologies. Therefore, understanding and quantifying paraspinal muscle activity is crucial for accurate diagnosis, treatment planning, and monitoring therapeutic interventions. This review discusses the clinical applications of sEMG in paraspinal muscles, including the assessment of low back pain, spinal disorders, and rehabilitation interventions. It explores how sEMG can aid in diagnosing the potential causes of low back pain and monitoring the effectiveness of physical therapy, spinal manipulative therapy, and exercise protocols. It also discusses emerging technologies and advancements in sEMG techniques that aim to enhance the accuracy and reliability of paraspinal muscle assessment. In summary, the application of sEMG in paraspinal muscles provides valuable insights into muscle function, dysfunction, and therapeutic interventions. By examining the literature on sEMG in paraspinal muscles, this review offers a comprehensive understanding of the current state of research, identifies knowledge gaps, and suggests future directions for optimizing the use of sEMG in assessing paraspinal muscle activity.

## 1. Introduction

The worldwide prevalence of musculoskeletal disorders has reached approximately 1.3 billion cases in which low back pain (LBP) has accounted for a high proportion of prevalent cases, representing 36.8% of the total [1]. With the expanding and aging population, the burden of LBP is particularly pronounced in low-income and middle-income countries, where the rapid demographic changes outpace the availability of resources to effectively address the issue, thus intensifying the effects in these regions [2].

The paraspinal muscles play a crucial role in maintaining spinal stability, posture, and movement, making them a topic of significant interest and investigation in various fields, including biomechanics, rehabilitation, and sports medicine [3,4,5,6,7]. It was observed that there were some differences in paraspinal muscles among the individuals with LBP, such as the change in cross-sectional surface area and fat infiltration [8,9].

Surface electromyography (sEMG), a non-invasive technique that captures and analyzes the electrical activity of muscles, has emerged as a powerful tool for investigating muscle function [10,11]. With its ability to measure the electrical signals generated by motor unit activity and the recruitment of muscle fibers, sEMG offers valuable insights into the mechanisms underlying muscle contraction and neuromuscular control [6,12,13,14]. It is utilized not only in healthy populations to evaluate the function and interactions of muscles during functional activities, sports, and exercise but also in clinical populations to comprehend muscle adaptations, dysfunctions, and impairments associated with musculoskeletal injuries, pain, and pathological conditions [15,16,17,18,19,20,21,22,23]. Furthermore, sEMG presents novel prospects in the control of robotic devices, prostheses, and exoskeletons. The utilization of sEMG signals to control robotic systems has been a long-standing concept. Presently, numerous devices have achieved the functionality of converting operator sEMG signals into force, and there have been advancements in the realization of real-time myoelectric signal control in exoskeleton robotics [24,25].

Traditionally, the functional investigation of paraspinal muscles has relied on invasive procedures, such as needle EMG and muscle biopsies, which limit their widespread utilization and pose inherent risks [26,27]. However, sEMG has provided new methods of paraspinal muscle research by enabling non-invasive, dynamic assessments. Arokoski et al. measured intramuscular and sEMG while assessing the activity of the paraspinal muscles and concluded that sEMG measurements could be used to evaluate the function of the multifidus muscles [28]. The versatility of sEMG allows for the assessment of paraspinal muscle activity during various tasks, such as static postures, dynamic movements, and specific exercises, facilitating a comprehensive understanding of their role in daily activities and pathological conditions. In recent years, the application of sEMG has gained significant traction, shedding light on various musculoskeletal conditions, optimizing sports performance, and guiding rehabilitation strategies [18,29,30,31,32,33,34]. We conducted a search on PubMed for relevant literature regarding the application of surface electromyography (sEMG) in assessing paraspinal muscles. The search was conducted up to 14 July 2023, using the following search strategy: (“sEMG”[All Fields] OR (“electromyography”[MeSH Terms] OR “electromyography”[All Fields] OR (“surface”[All Fields] AND “electromyography”[All Fields]) OR “surface electromyography”[All Fields])) AND (“paraspinal muscles”[MeSH Terms] OR (“paraspinal”[All Fields] AND “muscles”[All Fields]) OR “paraspinal muscles”[All Fields]). A total of 831 articles were retrieved. After screening titles, abstracts, and conclusion sections, 219 articles were selected. Subsequently, through further examination of the literature content, 153 relevant articles were identified.

## 2. Paraspinal Muscles

The paraspinal musculature assumes a fundamental role in spinal extension and contributes substantively to spinal stabilization. Its anatomical composition delineates into three distinct categories: the anterior ensemble, the lateral assembly, and the posterior consortium. The origins and terminations of the paraspinal muscles are relatively fixed, yet the specific anatomical details may exhibit slight variations due to factors such as anatomical location, physiological status, and ethnic differences. These muscles, connected by tendons, fascia, and other tissues originating from vertebral bodies, vertebral arches, and transverse processes, contribute to the support and stability of the spinal column. Additionally, they actively participate in various movements and postural adjustments of the spine [35,36,37,38,39,40].

Among these muscular entities, particular emphasis is placed on the multifidus muscle and erector spinae concerning spinal movement and stabilization. The multifidus muscle, a deep-seated muscle group located on both sides of the spine, primarily contributes to spinal stability and movement. The origin of the multifidus is at the vertebral arches and fascia above them. In the lumbar, thoracic, and cervical regions, the muscle fibers of the multifidus originate from ligaments or fascia above the vertebral arches and transverse processes. The fibers of the muscle incline inward and terminate at the corresponding vertebrae, intervertebral discs, and fascia between the vertebral bodies and arches. In the lumbar region, the muscle fibers and tendons of the multifidus typically conclude at the fascia above the next level of vertebrae and intervertebral discs. The multifidus muscle provides stable support to the spine and is involved in various movements and postural adjustments of the vertebral column [35,36,37,38]. The erector spinae muscles constitute a group of muscles located on either side of the spine, primarily responsible for providing support and stability to the spinal column. They also play a role in various movements, including extension, rotation, and lateral bending of the body. The upper origin of the erector spinae is situated at the external occipital protuberance, transverse processes of the cervical vertebrae, transverse processes of the thoracic vertebrae, and the fascia above the lumbar vertebrae. The muscle fibers descend along the spine, converging into three bundles near the lumbar vertebral arches. The lower termination divides into two bundles, with the spinalis thoracis terminating at the intercostal muscles and ribs, and the iliocostalis lumborum ending at the fascia between the ribs and iliac crest [37,39,40].

## 3. sEMG

sEMG signal processing techniques play a crucial role in extracting meaningful information from the recorded electrical activity of muscles. Among these techniques, two widely used methods are time domain analysis and frequency domain analysis, which provide valuable insights into muscle function and performance.

Time domain analysis involves examining the temporal characteristics of sEMG signals. There are two frequently used parameters, integral electromyography (IEMG) and root mean square (RMS), which contribute to the comprehensive evaluation of muscle function. The IEMG, obtained by integrating the rectified sEMG signal, quantifies the overall muscle activation during a specific time period [41,42]. The IEMG reflects the cumulative electrical activity produced by active motor units and is often used to assess muscle force production and fatigue [43]. IEMG activity tends to increase with force generation in various muscles. However, during muscle fatigue, the relationship between IEMG activity and force becomes disconnected or dissociated [42,44,45]. The RMS is a fundamental parameter used in sEMG analysis, representing the magnitude of electrical activity recorded from the muscle. It quantifies the overall amplitude of the sEMG signal and provides insights into muscle activation levels and force production. The electrophysiological monitoring of skeletal muscle fatiguability can be achieved by analyzing alterations in spectral parameters of the EMG signal achieved through fast Fourier transformation. This technique enables the tracking of alterations in the median frequency of the power spectrum and the RMS value of the entire signal during fatiguing activities [46,47]. 

Frequency domain analysis involves examining the spectral characteristics of sEMG signals. By employing Fourier transform or other spectral analysis techniques, researchers can explore the frequency components of the signals and derive informative parameters related to muscle function. Mean power frequency (MPF) represents the average frequency weighted by the power spectrum of the sEMG signal. It provides insights into the frequency characteristics of muscle activity and is associated with muscle fiber type distribution and fatigue. Median frequency (MF) represents the frequency below which 50% of the total power spectrum lies. It offers valuable information about the spectral distribution of muscle activity. The MF is less prone to noise interference and exhibits greater sensitivity to muscle fatigue-related changes during voluntary contractions [48]. There seems to be a correlation between the decline in median frequency slope induced by fatigue and the distribution of underlying muscle fiber types I and II [49]. During muscle fatigue, there is a decline in MF values due to changes in the recruitment and firing rates of motor units. Mannion et al. investigated the correlation between EMG signs of fatigue and the duration of isometric contraction of the back extensors to fatigue. It was found that endurance is restricted by the most easily fatigued section of the muscle group. The MF of the EMG power spectrum is a suitable method for monitoring fatigue in the back muscles [50]. By monitoring MF, researchers and clinicians can gain valuable information about the onset and progression of muscle fatigue, as well as the adaptability of neuromuscular control strategies. Kankaanpää et al. developed a back muscle endurance test based on the power spectral indices and endurance time of paraspinal muscles’ sEMG signal. The study indicated that paraspinal muscle spectral indices, MF and MPF, measured were good predictors of endurance time before the onset of total muscle fatigue [51]. Ebenbichler et al. aimed to investigate the efficacy of the MF sEMG method in assessing back muscle fatigue in neuromuscular and muscle metabolic functions in individuals with CHRONIC LBP. The study showed that MF suggested that this method held considerable potential as a prognostic and diagnostic biomarker capable of identifying early indicators or signs of an accelerated aging process within the neuromuscular system [52].

## 4. Evaluate Paraspinal Muscles with sEMG

sEMG finds extensive application in the assessment of paraspinal muscles. It is employed in the realm of rehabilitation to formulate personalized rehabilitation plans, optimizing the strength and coordination of paraspinal muscles. Additionally, sEMG is widely utilized in the diagnosis and monitoring of paraspinal muscle function, aiding in understanding aspects such as spinal stability, posture control, and movement coordination. The application of sEMG in assessing paraspinal muscles not only provides a novel perspective for clinical diagnosis but also offers robust support for individualized rehabilitation and treatment (Figure 1).

### 4.1. sEMG in Health and Behaviors

Several studies have conducted measurements of paraspinal muscle sEMG activity in various occupational settings, providing guidance for the improvement of work-related muscle strain. Anton et al. evaluated the effect of lift teams on kinematics and muscle activity among bricklayers [15]. Lyons evaluated EMG differences of muscle activation across three commonly performed kettlebell exercises [53]. Jakobsen et al. measured muscular activity of the lower back throughout the working day using sEMG among blue-collar workers with manual lifting tasks [54]. Lee et al. compared trunk biomechanics during barbell back squats in three-foot postures among weight lifters [55]. Forrest et al. evaluated the muscle activation of lumbar spinal muscles among fast bowlers [56]. Kawahara et al. assessed the correlation between working postures while holding a chain-saw and paraspinal muscle activities with sEMG [16]. Yoo et al. investigated the influence of different lifting and lowering heights to back muscle activity during manual material handling [57]. Yoo et al. compared different seat supports among visual display terminal workers during their working at computers, which concluded that an unstable cushion-ball seat support might prevent work-related neck disorders [58]. Schmid et al. evaluated the influence of different sling-based infant carrying techniques. The study demonstrated that carrying an infant alternating on both sides with a sling could be beneficial for preventing musculoskeletal pain in women after childbirth [59].

Therapeutic exercises have been found to effectively activate the paraspinal muscles. Arokoski et al. evaluated the activities of the paraspinal and abdominal muscles during various therapeutic exercises by sEMG. It was found that simple therapeutic exercises were effective in activating both the abdominal and paraspinal muscles [60]. Arokoski et al. determined that therapeutic exercises performed without external assistance exhibit greater efficacy in activating the lumbar paraspinal muscles compared to exercises assisted manually [61]. Khosrokiani et al. assessed the influence of therapeutic exercises to paraspinal muscles. The result suggested that the use of back bridge exercises strengthened the multifidus muscles [62]. Mello et al. evaluated the EMG activity of the lumbar multifidus and erector spinae muscles, in both the time and frequency domains, while performing the back bridge exercises. The study demonstrated that activation of muscles at a threshold of 30% is deemed adequate for maintaining lumbar stability and is also considered suitable for enhancing muscular endurance [63].

Walking under some conditions may potentially be beneficial for exercising the paraspinal muscles. Lee et al. used sEMG to measure the activation of the paraspinal muscles, specifically the multifidus and erector spinae, at various walking velocities and slopes. The study revealed that fast-paced walking exercises result in greater activation of the lumbar multifidus muscles compared to slow-paced walking exercises [17]. Zoffoli et al. examined the areas and patterns of activity of the external oblique, erector spinae longissimus, and multifidus muscles while performing walking and pole walking at varying speeds and inclinations by sEMG and observed a similar phenomenon [29]. Masumoto et al. compared the muscle activities of subjects while walking forward or backward in water, with and without water current. The results showed that the muscle activities of the paraspinal muscles were significantly higher when walking backward compared to walking forward [64]. 

Numerous exercise modalities have been proven to enhance the activation of paraspinal muscles, and the diverse effects of specific details during exercise on these muscles can be observed. This serves to guide patients with impaired paraspinal muscle function in engaging in more effective workouts. Colado et al. assessed the EMG activity of various paraspinal muscles during localized stabilizing exercises and multijoint or global stabilizing exercises. The finding indicated that among asymptomatic young individuals engaging in free-weight multijoint exercises, such as deadlifts performed at 70% of their maximum voluntary isometric contraction, lead to greater peak and mean levels of muscle activation compared to localized stabilization exercises with or without instability, as well as global exercises that incorporate localized stability [65]. Matthijs et al. examined the co-contraction of the superficial multifidus muscle at the lower lumbar spine during volitional preemptive abdominal contraction. The analysis of results indicates that utilizing volitional preemptive abdominal contraction strategies was an effective method for co-activating the superficial multifidus muscle, which might help to improve spinal protection and rehabilitation outcomes [66]. Park examined the impact of hand and knee positions on muscular activity during back extension exercises utilizing the Roman chair. The findings of this study suggested that altering the position of the knees and hands during back extension exercises using the Roman chair could result in varying levels of activation of specific muscles [67]. Wagner et al. measured and compared the mean activation of trunk muscles during laughter yoga to that of crunches and back lifting exercises. The finding suggested that laughter yoga might have a positive impact on both the quantitative and qualitative activation of trunk muscles [68]. Andrade et al. examined the impact of Pilates principles on the EMG activity of the paraspinal muscles when performing exercises on both stable and unstable surfaces. The results of this study demonstrated that the EMG activity of the iliocostalis muscle was greater during stable surface exercises utilizing Pilates principles. Conversely, the lumbar multifidus muscle exhibited the highest level of activation during stable surface exercises, regardless of the use of Pilates principles [69]. Kim et al. analyzed the selective EMG activity of the lumbar paraspinal muscles in both healthy male and female subjects while performing the prone trunk extension and four-point kneeling arm and leg lift exercises. The findings of this study suggested that the four-point kneeling arm and leg lift exercise was both safe and effective in selectively activating the lumbar multifidus muscle [70]. Masaki et al. analyzed the activity of the back muscles during quadruped upper and lower extremity lifts while varying the lifting direction and weight loading of the extremities. The result indicated that quadruped upper and lower extremity lift exercises involving shoulder and hip abduction were more effective in selectively strengthening the lumbar multifidus muscle on the side where the lower extremity is lifted [71]. Youdas et al. compared the level of sEMG activation in the paraspinal muscles during four different planking procedures. The findings of this study suggest that prone plank on ball with hip extension was able to elicit significant activation of the lumbar multifidus at a strengthening level [72]. Escamilla et al. evaluated the comprehensive activation of core musculature during exercises performed in the prone position, contrasting them with exercises executed. The study found that exercises conducted in a side position were more effective for recruiting the oblique and lumbar paraspinal muscles [73]. Calatayud et al. assessed whether utilizing a suspended modality leads to heightened trunk muscle activation during unilateral or bilateral isometric supine planks. The outcomes of the study suggested that incorporating unilateral variations with a suspended support yield the most substantial activation of the lumbar erector spinae muscles [74].

Appropriate resistance training can aid in increasing paraspinal muscle activation during exercise, which may have a positive impact on the functional conditioning of these muscles. Du Rose et al. examined the associations between the range of motion of the lumbar intervertebral joints and the activity of the paraspinal muscles during weight-bearing flexion in healthy individuals, utilizing quantitative fluoroscopy and sEMG. The study revealed modest to substantial yet noteworthy links between the modifications and ratios of muscle activity and the maximum range of motion of the intervertebral joints at different levels [18]. Nijem et al. examined the impact of variable resistance applied by chains on sEMG activity in the gluteus maximus, erector spinae, and vastus lateralis muscles during deadlifts. The finding of the study indicated that incorporating chain resistance during deadlifting could modify the muscle activation patterns of the lift [75]. Roth et al. evaluated that the activation patterns of trunk muscles vary depending on the barbell position in squat exercises with low weights in beginners. The study found that squat exercises with the barbell in the back, front, and overhead positions resulted in different activation patterns of the trunk muscles, which concluded that these exercises can be useful in activating the trunk muscles [76]. Nevertheless, excessive load during exercise can potentially lead to injury of the paraspinal muscles. Ma et al. evaluated the effect of a large lifting load and found that a large lifting load could lead to more spasms and more muscle activations on the erector spinae muscle in the relaxation period, which might lead to low back disorders [77].

Whole body vibration training was a popular exercise method for athletes and patients. Seroussi et al. discerned a markedly greater magnitude of both average and peak-to-peak estimated torque across nearly all frequency spectra when subjects engaged in seated activities involving vibration, as opposed to static sitting, as assessed through sEMG [78]. The increase in muscle activity caused by whole body vibration was significant, particularly for the back muscles, which was up to 19.0% MVC [79]. However, Nolan et al. indicated that a person was at an increased risk for low back injury when their back muscles were fatigued and they were exposed to whole body vibration [80].

### 4.2. sEMG in Lumbar Disorders

#### 4.2.1. LBP

The sEMG signals of the paraspinal muscles in patients with LBP differ from those of healthy individuals. Lu et al. compared the muscle activity strategies between healthy persons and patients with LBP. The study demonstrated that there was a difference in muscle activity patterns between healthy persons and patients with LBP [81]. Cram et al. assessed the patterns of sEMG activity in the patients with LBP, which demonstrated that the differences between both sides’ sEMG activity in the lumbar and cervical paraspinal muscle groups were important to distinguish between headache patients and LBP patients [4]. Cooper et al. combined the sEMG with a standardized isometric fatigue test to investigate the mechanisms of chronic LBP. The study proposed that the muscle excess fatigue might be peripheral in origin [82]. Furthermore, there is a good reproducibility for evaluating patients with LBP by sEMG. Lee et al. measured the repeatability of sEMG in the LBP and healthy subjects, which showed a great repeatability [45].

Changes in sEMG signals may be associated with morphological and histological alterations in the paraspinal muscles. Djordjevic et al. found a correlation between the sEMG signal and the relative thickness change in the transversus abdominis during muscle activation in individuals, both with and without LBP [83]. Crossman et al. investigated the relationship between the excessive paraspinal muscle fatigue in subjects with chronic LBP and muscle fiber content. The study documented there was non-existent histomorphometric discrepancy between the subjects with and without chronic LBP [84]. 

sEMG has the capacity to elucidate the functional status of paraspinal muscles in individuals afflicted with LBP. Becker et al. evaluated the activation of lumbar erector spinae in the patients with specific LBP during activities of daily living by sEMG. The study determined there was a high relationship in lumbar erector spinae between the changes of activity and the degree of function [19]. Chiou et al. investigated the relationship between the frequency characteristics of sEMG and the functions of paraspinal muscles in the subjects with LBP. The findings demonstrated alterations in the sEMG profile and its association with LBP disability, which determined that spectral characteristics of sEMG reflect muscle function [85]. Jalovaara et al. evaluated the sEMG of paraspinal muscles among simple LBP patients and the LBP patients with and without verified LDH. The study showed that sEMG was a valid tool for indirectly assessing pain in LBP patients [86].

Patients with LBP exhibit different changes in paraspinal muscle activity during different activities, which can be distinguished from other healthy participants. Yang conducted a comparative investigation which compared the activity of lumbar multifidus muscles in the individuals with and without LBP during static stoop lift. The study demonstrated that less activity of lumbar multifidus muscles observed in the subjects with LBP might contribute to decrease the lumbar stabilization during stoop lift [87]. Da Silva et al. assessed the activity of trunk muscles in the subjects with and without LBP during a one-legged stance task. The research proved that there was less lumbar muscle activity and more coactivation between rectus adominis and multifidus muscles in the patients with LBP during the one-legged stance task compared with their healthy counterparts [88]. Sanderson et al. generated a topographical map of the EMG amplitude by high-density sEMG in the patients with and without LBP during a lumbar extension endurance task. The study demonstrated that individuals without symptoms exhibited a spatial redistribution of lumbar erector spinae muscle activity during an endurance task, whereas this adaptation was diminished in the subjects with LBP [89]. Shah compared the EMG activity between the individuals with and without chronic LBP and found there was a significant increase in the recruitment of the lumbar multifidus muscle with increased lumbar lordosis in patients with chronic LBP during quadruped exercise [90]. Varrecchia et al. elucidated distinct motor strategies utilized by individuals with and without LBP during fatiguing frequency-dependent lifting tasks, through the utilization of muscle coactivation parameters. The study demonstrated the possibility of discerning distinct motor strategies between individuals with and without LBP, which showed that the individuals with LBP engage in greater coactivation of their trunk muscles more than the healthy controls, via the implementation of a fatiguing trunk-stiffening strategy [91]. Sánchez-Zuriaga et al. assessed the lumbopelvic motion and erector spinae activity patterns during trunk flexion–extension movements in patients with recurrent LBP during pain-free periods and matched asymptomatic individuals. The finding suggested that patients with a history of LBP during pain-free periods display slight changes in EMG activity patterns as well as reduced maximum lumbar flexion ranges, distinguishing them from non-LBP subjects [92]. Kuriyama et al. compared the kinesiologic sEMG in the patients with LBP and healthy subjects during different trunk motions. The study showed that continuous muscle activity and no intermuscular time lag were respectively observed in the LBP patients in the full trunk flexion position and on axial rotation, which was contrary to the healthy controls. This suggested the paraspinal muscles played a role in spinal stabilization [93]. Arvanitidis et al. measured the correlation between spatial oscillations in sEMG activity and trunk-extension torque in individuals, distinguishing between those with and without chronic LBP. The study suggests that patients with chronic LBP are unable to augment the shared fluctuations in torque and high-density sEMG activity during increased higher lumbar extension forces [94]. 

sEMG characteristics of the paraspinal muscles can aid in distinguishing different types of LBP. Balasch-Bernat et al. suggested that patients with continuous chronic LBP exhibited elevated EMG activity in the erector spinae and multifidus muscles during the isometric and concentric phases of back extension exercises, as compared to healthy individuals. This distinction in muscle activity was also present to a lesser degree when compared to patients with recurrent LBP [95]. Suehiro et al. assessed the pattern of trunk muscle activation during the active hip abduction test in the patients with recurrent LBP. The study indicated that in individuals with recurrent LBP, the activation of contralateral erector spinae occurred with a delay compared to those without recurrent LBP during both right and left active hip abduction tests. Additionally, during the left active hip abduction test, the onset of ipsilateral erector spinae activation was present later in individuals with recurrent LBP compared to those without recurrent LBP [96]. Humphrey et al. investigated different sEMG parameters among the healthy, chronic LBP, and past history subjects with LBP. It was observed that there were statistically significant differences in the sEMG variables between chronic LBP patients and normal controls, in which half-width, initial median frequency, and peak amplitude were great parameters to distinguish the individuals with and without chronic LBP [97]. Cassisi et al. analyzed the activity of lumbar paraspinal muscles in patients with chronic LBP during isometric exercise and rest. The study documented that maximum surface integrated EMG could be helpful for the classification of chronic LBP during isometric exercise [98].

sEMG can be employed to evaluate the functional state of paraspinal muscles in individuals with LBP attributed to varying professions and lifestyle habits. Adeyemi et al. evaluated the sEMG activities of erector spinae and trapezius muscles among schoolchildren from carrying backpacks of different weight. The study indicated that the activity of paraspinal muscles in the lower back was more influenced by backpack carriage than the upper back among schoolchildren [99]. Martinez-Valdes et al. evaluated the spatial distribution of the activities of the erector spinae by high-density EMG in rowers with and without LBP. The study found higher amplitude and less complexity in the subjects with LBP, which documented that the magnitude of activation and the distribution of erector spinae activity were observed changed in the LBP group [100]. Hao et al. scrutinized the characteristics of sEMG in lumbar muscles during prolonged contractions in military personnel, both with and without chronic LBP. The study indicated that the non-uniform spatial distribution and asymmetry in lumbar muscle activity constituted noteworthy factors in individuals experiencing chronic LBP [101]. Zou et al. examined the impact of core stability training on the management of nonspecific LBP among nurses, which suggested that core stability training had the potential to ameliorate symptoms, enhance the fatigue resistance of core muscles, and promote the balance of bilateral multifidus muscle function in patients suffering from nonspecific LBP [21].

The utilization of the flexion–relaxation ratio (FRR) to differentiate paraspinal muscle activity between individuals with LBP and healthy individuals is considered effective. Watson et al. developed the flexion–relaxation ratio to reflect the change in paraspinal muscle in the patients with chronic LBP and the healthy controls. The study demonstrated that the measurement of FRR was a reliable way to identify dynamic sEMG activity of the paraspinal muscles as well as being useful in distinguishing chronic LBP patients and healthy subjects [102]. Gouteron et al. evaluated the erector spinae longissimus and multifidus muscles of nonspecific chronic LBP patients with sEMG and found that all methods used to calculate FRR for erector spinae longissimus and multifidus muscles could be great methods for identifying the flexion–relaxation phenomenon (FRP) in nonspecific chronic LBP patients [103]. 

The flexion–relaxation ratio can also be employed to evaluate the functional status of the paraspinal muscles. Owens et al. investigated the FRR of paraspinal muscles in patients with back-related leg pain. The study demonstrated that the patients with back-related leg pain who showed a very low FRR suffered more disability, more clinical findings, and decreased motion [104]. Mak et al. compared the patients with LBP before and after rehabilitation treatment with their healthy counterparts. The study suggested that FRR in sitting in normal subjects was significantly greater than LBP patients. And the FRR increased in sitting in LBP patients after rehabilitation [105]. Kankaanpää et al. compared the lumbar paraspinal muscles’ fatigability between chronic LBP patients and healthy controls during an isometric endurance task. The study demonstrated that the fatigability for chronic LBP patients was faster than the healthy counterparts during an isometric back extension endurance task [106]. Shigetoh et al. evaluated diminished lumbar flexion–relaxation and decreased variability in muscle activity distribution among patients with chronic LBP. The study concluded that integrating pain-related factors with the assessment of FRR and muscle variability might enhance the ability to predict chronic LBP disability [107]. The relationship between asymmetric FRP of lumbar muscles and trunk lateral range of motion in the patients with nonspecific chronic LBP was compared and a moderate correlation was found between them [108] (Table 1).

#### 4.2.2. LDH

The sEMG signals of the paraspinal muscles in patients with lumbar disc herniation reflect a decrease in muscle strength and an increase in fatigue. Zhao compared the average myoelectric amplitude of sEMG between the patients with LDH and their healthy counterparts. The study investigated that there was an imbalance in myoelectric activity in the individuals with chronic LDH, and the muscle strength on the affected side was significantly reduced [23]. Ramos assessed lumbar multifidus fatigue in the patients with LDH associated with LBP and found increased fatigue of the lumbar multifidus compared with their control counterparts [109]. Leinonen et al. investigated the lumbar paraspinal muscle reflexes during sudden upper limb loading in patients with sciatica and indicated that chronic pain could impair lumbar feedforward control in chronic LBP patients [110]. Following percutaneous endoscopic lumbar discectomy, the condition of the paraspinal muscles in LDH patients improved. Li et al. assessed the muscle functions of patients suffered LDH after percutaneous endoscopic lumbar discectomy by sEMG. The study demonstrated that percutaneous endoscopic lumbar discectomy for individuals with LDH could normalize paraspinal muscle activation during lumbar flexion–extension movement [111].

#### 4.2.3. Lumbar Spinal Stenosis (LSS)

The sEMG characteristics of the paraspinal muscles in patients with LSS differed from those in healthy individuals, but the endurance of these muscles was good. Nüesch et al. evaluated the muscle activation of the erector spinae and multifidus using sEMG in the patients with LSS. The study demonstrated that during midstance, higher activation of the multifidus and erector spinae showed in the patients with LSS compared with the healthy controls [112]. Leinonen et al. assessed paraspinal muscle activity by sEMG during trunk flexion–extension movement and muscle endurance during the dynamic isoinertial back endurance test. The study documented that the endurance of paraspinal muscles in LSS patients was good [113].

The assessment of paraspinal muscle sEMG in postoperative LSS patients can aid in the evaluation of changes in the paraspinal muscles. Kääriäinen et al. evaluated the responses of paraspinal muscles in the patients with LSS before and after surgery during sudden upper limb loading by sEMG. The activation of paraspinal muscles could be observed in the patients with LSS, which indicated an extensive loss of motor functions in the patients with LSS. Although paraspinal muscle activation profiles were unchanged after surgery, during two years of follow-up, muscle activation profiles tended to further deteriorate [114,115] (Table 2).

### 4.3. sEMG in Cervical and Thoracic Disorders

#### 4.3.1. Adolescent Idiopathic Scoliosis (AIS)

There are discernible differences in the sEMG characteristics of the paraspinal muscles between individuals with and without AIS. Furthermore, different types of scoliotic curvatures are associated with distinctive features in the EMG signals of the paraspinal muscles. Yuan et al. retrospected the female patients with nonspecific LBP with and without lumbar scoliosis. The study demonstrated the posterior muscles, the ratio of RMS on paraspinal muscles, and relaxation time during the FRP in the scoliotic patients were found greater than non-scoliotic patients, which showed the differences of paraspinal muscles in nonspecific LBP patients with or without lumbar scoliosis [116]. A single-center prospective cohort study investigated the correction between the sEMG activity according to the scoliosis curve type and AIS curve types. The study demonstrated a different characteristic of sEMG in different types of adolescent idiopathic scoliosis curves [117]. Gaudreault et al. compared the L5/S1 moments during isometric efforts in extension in the patients with idiopathic scoliosis with their healthy counterparts and argued that there might be a potentially compensatory mechanism at the lower level of the spine [7]. The sEMG characteristic of paraspinal muscles between the patients with scoliosis before and after spine fusion and their healthy counterparts was compared in a prospective study. The study demonstrated that unbalanced sEMG activity in the paravertebral muscles could be found in the patients with AIS, which could be decreased after spine fusion. Less and higher sEMG activities were separately found in the thoracic and lumbar paraspinal muscles after spine fusion [118]. A prospective study investigated that there was a significant potential to the evaluation and treatment of idiopathic scoliosis by combining the analysis of the spinal growth velocity and EMG ratio [119].

sEMG can be utilized to detect asymmetrical EMG activity in the paraspinal muscles on both sides of the spine in adolescents with AIS, which may provide guidance for the diagnosis and treatment of patients with AIS. Chan et al. compared the influence of fatigability of paraspinal muscles in the patients with AIS by EMG, and found there was a higher erector spinae activity at the convex side of AIS. The study demonstrated that an imbalance in paraspinal muscles could play a potential role in the rehabilitation for AIS patients [120]. Cheung et al. evaluated the relationship between the sEMG of paraspinal muscles and the progression of the scoliotic curve. The study demonstrated that sEMG of the paraspinal muscles could forecast the progression in idiopathic scoliosis [121]. Liang et al. further investigated the sEMG signals of paraspinal muscles in the patients with AIS and found a similar conclusion that the dynamic asymmetric of the erector spinae group of muscles could predict scoliosis aside [122]. Cheung et al. evaluated asymmetry in paraspinal muscle activities in the patients with AIS after sEMG biofeedback posture training and found that sEMG biofeedback posture training could reduce the asymmetric paraspinal muscle activities and the curve progression [123]. Tsai et al. combined the isokinetic back system with quantitative sEMG to evaluate paraspinal muscle activities. The study documented that during isokinetic flexion and extension exercises, the sEMG activities of the thoracic muscle were significantly higher on the concave side than on the convex side in AIS patients with larger curves, which was different from the healthy control group and those with AIS with smaller curves [124].

Different exercise modalities can either promote or inhibit the activation of paraspinal muscles in individuals with AIS, providing a reference for selecting appropriate functional exercise methods for AIS patients. Ko et al. evaluated the muscular activation patterns in the bilateral erector spinae by sEMG. They found that using asymmetric spinal stabilization exercise can improve the severity of scoliosis, especially at the concave side of paraspinal muscles [125]. Mooney et al. treated patients with AIS by a resistive training program for torso rotation, in which sEMG showed inhibition of lumbar paraspinal muscles [126]. He et al. assessed the activation of paraspinal muscles among the patients with AIS before, during, and after the Schroth exercise by sEMG. The study suggested that the asymmetric and symmetric exercises elicited greater sEMG activity on the convex and concave side, respectively. Additionally, weight-bearing exercises resulted in increased paraspinal muscle contractions on both sides of the scoliotic curve [127]. Farahpour et al. evaluated the sEMG response between the patients with AIS and healthy individuals to dynamic postural perturbation. The study indicated that asymmetry of muscle activity in AIS patients relies on the direction of the perturbation [128]. Perret et al. confirmed the presence of short-latency responses and later activities in sEMG after a postural perturbation [129] (Table 3).

#### 4.3.2. Spinal Cord Injury (SCI)

Measuring sEMG responses of paraspinal muscles in SCI patients may provide a novel method for evaluating the functional status of the spinal cord in these patients. Singh et al. recorded sEMG from trunk muscles during uncompensated sitting between the children with spinal cord injury and their healthy counterparts. The study demonstrated significantly higher partial thoracic and lumbar paraspinal muscle activation, which indicated the residual influence on paraspinal muscles of spinal motor circuitry supplying these postural muscles [130]. Hoglund et al. evaluated the effect of epidural spinal cord stimulation on the patients with SCI by paraspinal muscle sEMG. The study found that different stimulations result in different responses of different paraspinal muscles, which could be helpful for tailoring targeting of those muscle groups in this study [131].

#### 4.3.3. Neck Pain (NP)

Patients with NP exhibit abnormal sEMG activation in their paraspinal muscles. Sremakaew et al. compared the paraspinal muscle activity in the patients with NP and the healthy individuals, and found that except for the upper trapezius, there was higher activity in all muscles in the NP group [132]. Lecompte et al. compared the sEMG changes of paraspinal muscles among the fighter pilots with and without NP and the healthy non-pilot subjects. The study suggested that there was a difference in the muscles’ function in the coronal plane between the fighter pilots with and without NP [133]. Park et al. compared the activation of the bilateral cervical paraspinal muscles between the patients with unilateral posterior NP and their healthy counterparts during prone neck extension. The study demonstrated that unilateral posterior NP significantly alters neck motion and muscle activation during active neck extension [134]. Traction and local functional exercises may be beneficial in the treatment of patients with NP. Nanno et al. evaluated the effects of cervical intermittent traction on the patients with NP. The study demonstrated that the cervical intermittent traction could contribute to relieve pain, increase the frequency of EMG signals and improve blood flow in affected muscles [135].

#### 4.3.4. Whiplash-Associated Disorder (WAD)

The utilization of sEMG in assessing patients with WAD provides a novel perspective on the risk factors associated with this type of injury. Siegmund et al. determined the neck muscle response during awareness of the presence and timing of whiplash-like perturbation affects. The study demonstrated that surprised females showed larger retractions, which might produce larger tissue strains and potentially increase injury [20]. In the patients experiencing sequential whiplash-like perturbations, significant changes were observed in paraspinal muscle variables, which indicated habituation might be a potential confounding factor in studies investigating whiplash injuries through repeated perturbations [136]. Mang et al. investigated patients who experienced rear-end perturbations with a loud preimpact tone and the response of the cervical multifidus muscle. The study demonstrated that a loud preimpact tone might reduce the strain in the cervical facet capsule, which manifested as the decrease in peak C6 multifidus activity and head kinematic responses, which might reduce the risk of whiplash injury during rear-end collisions [137]. Descarreaux et al. evaluated the function of paraspinal muscles between the patients with WAD and healthy controls. The study indicated that the patients with WAD were able to produce isometric forces with spatial precision in which the time to peak force was increased [138] (Table 4).

## 5. sEMG in Massage Therapy and Rehabilitation Training

Massage therapy can promote the normalization of abnormal paraspinal muscle EMG activity in patients with LBP and aid in the relief of pain. Colloca et al. prospectively analyzed the sEMG reflex responses during the mechanical force, manually assisted spinal manipulative therapy in the patients with LBP. The study identified the neuromuscular characteristics in the patients with LBP and demonstrated that mechanical force, manually assisted spinal manipulative therapy resulted in consistent and generally localized sEMG responses in patients with LBP [139]. Daneau et al. researched the clinical changes of muscle fatigue after massage in nonspecific chronic LBP individuals. The results of this study revealed a decrease in pain severity in individuals with chronic LBP and localized muscle fatigue following a single 30 min massage session [140]. Bicalho et al. investigated the instant effects of high-velocity spine manipulation on paraspinal activity while performing flexion–extension trunk movements. The findings showed that a high-velocity spinal manipulation technique induces acute changes in the EMG activity while executing flexion–extension movements in individuals suffering from chronic LBP [141]. Qiao et al. evaluated the relationship between the alteration of sEMG and symptom relief before and after massage therapy in the patients with acute nonspecific LBP and healthy controls. The study suggested that symptoms were alleviated and myoelectric activity of the paraspinal muscles were normalized after massage therapy in the acute nonspecific LBP patients [5].

There is still controversy regarding the efficacy of rehabilitation programs in the treatment of patients with LBP. Certain findings suggest that exercise modifies the activation and recruitment of paraspinal muscles in individuals with LBP, thereby promoting functional recovery. Kim et al. investigated the short-term effects of direct vibration on the deep trunk muscles of patients with nonspecific chronic LBP. The study demonstrated that targeted and localized vibration stimulation applied during trunk muscle contractions via stabilization exercises brought about significant alterations in muscle activity patterns [22]. Suehiro et al. investigated whether a single session of abdominal drawing-in exercises could rectify the deviant activation patterns of trunk muscles observed during lifting tasks in individuals with recurrent LBP. The findings implied that abdominal drawing-in exercises might be efficacious in enhancing the recruitment pattern of muscles in individuals afflicted with recurrent LBP [6]. Masaki et al. examined the activities of back muscles and sagittal spinal alignment in young males affected by LBP with a history during the performance of quadrupedal unilateral limb extension exercises. The outcomes of this study propose that during quadrupedal unilateral limb extension exercises, there was an escalation in activity of the superficial muscles of the trunk, such as the latissimus dorsi and erector spinae muscles, in young males afflicted by LBP with a history [142]. Haładaj et al. assessed the patients with LBP after multiple impulse therapy and concluded that multiple impulse therapy was a non-invasive and efficacious method of treating LBP. Multiple impulse therapy substantially lessened paraspinal muscle tone, as substantiated by the results of sEMG, and exhibited a potent analgesic effect [143]. Kahlaee et al. compared the impact of abdominal hollowing and abdominal bracing techniques on the activity pattern of the paraspinal muscles during prone hip extension among individuals with or without nonspecific chronic LBP. The study showed that the group with LBP demonstrated elevated levels of muscle activity in all measured muscles. Furthermore, execution of the abdominal hollowing maneuver resulted in decreased amplitude of the erector spinae muscle activity for both groups [144].

However, other studies have not observed significant improvements in paraspinal muscle function among LBP patients following rehabilitation programs. Lewis et al. assessed the change in the activation of paraspinal muscles after a rehabilitation program in the individuals with chronic LBP. The study demonstrated that muscular activity did not exhibit a reduction subsequent to the administered treatment, which might be indicative of a transitional phase characterized by adaptation or alterations in motor control patterns [145]. De Souza Júnior evaluated the effects of Kinesio Taping on paraspinal muscles in women with LBP presenting fears and beliefs related to physical activity. The study determined that Kinesio Taping had an instantaneous impact on the peak torque of the erector spinae in women suffering from nonspecific chronic lower back pain, who demonstrated apprehension and beliefs associated with physical activity. However, there was no observed effect on muscle activity [146]. Arokoski et al. scrutinized the engagements of paraspinal musculature in the context of therapeutic exercises prescribed for individuals afflicted with nonspecific chronic LBP. In this study involving a limited cohort of patients with chronic LBP, it was observed that active physical rehabilitation did not have an impact on the activities of the abdominal and back muscles during therapeutic exercises that are commonly used for treating chronic LBP [147].

## 6. Limitations and Summary

sEMG, as a non-invasive bio signal detection technique, holds extensive potential in clinical applications. However, despite its significant utility in disease diagnosis, kinematic analysis, rehabilitation medicine, and efficacy feedback, it is not without limitations. Primarily, sEMG signals are susceptible to various interfering factors, including muscle fatigue, changes in body posture, and electrode placement on the skin. These interferences may exacerbate signal noise, thereby compromising the accuracy and reliability of data. Moreover, the acquisition of sEMG signals is constrained by the positioning of surface electrodes and the diversity of sensor types, potentially impeding the comprehensive capture of muscle activity information. Furthermore, interindividual physiological variances and variations in muscle anatomical structures may also impede the interpretation and comparison of sEMG signals. Thus, there is an urgent imperative to establish standardized models for sEMG detection. Despite the foundational knowledge of sEMG as a non-invasive muscle assessment technique dating back to the early 20th century, much of the developmental efforts have been decentralized among disparate scientific communities worldwide, leading to methodological diversity and a lack of standardization. Although the SENIAM (Surface Electromyography for the Non-Invasive Assessment of Muscles) project endeavors to standardize sEMG sensors, sensor placement procedures, and signal processing methods, a genuine international standard has yet to be established. Hence, standardization remains pivotal in propelling the widespread adoption of this technology [148].

The development and refinement of standardized protocols for sEMG data collection and analysis would greatly enhance the comparability and reproducibility of findings across studies. Establishing guidelines for electrode placement, signal processing, and normalization techniques would facilitate more robust and consistent measurements of paraspinal muscle activity. This standardization would enable more reliable comparisons across different populations, interventions, and research settings. Advancements in technology, such as wearable sEMG devices and wireless sensors, offer exciting opportunities for ambulatory monitoring and real-time assessment of paraspinal muscle function. These innovations could allow researchers and clinicians to capture muscle activity in naturalistic settings, providing valuable insights into daily activities, postural control, and dynamic movements. The application of machine learning algorithms and artificial intelligence techniques to sEMG data analysis could unlock new avenues for predicting and monitoring paraspinal muscle dysfunction. By leveraging large datasets and sophisticated algorithms, it may be possible to identify novel biomarkers, patterns, and predictors of LBP, leading to more personalized interventions and preventive strategies [122,149,150,151,152]. Lastly, translating sEMG research findings into clinical practice represents a crucial prospect. Integrating sEMG assessments into routine clinical evaluations, rehabilitation programs, and ergonomic interventions could enhance diagnostic accuracy, treatment efficacy, and injury prevention. Implementation of sEMG-based biofeedback systems could empower individuals to self-regulate their muscle activity, facilitating optimal movement patterns and reducing the risk of musculoskeletal disorders [109,123,153].

In conclusion, the body of research on sEMG and paraspinal muscles has shed light on the intricate relationship between muscle function and clinical applications. These research hold immense potential to deepen our understanding of muscle function, improve diagnostic and therapeutic approaches, and ultimately enhance the quality of life for individuals affected by paraspinal muscle-related conditions [4,5,86,98].

## Figures and Tables

**Figure 1 diagnostics-14-01086-f001:**
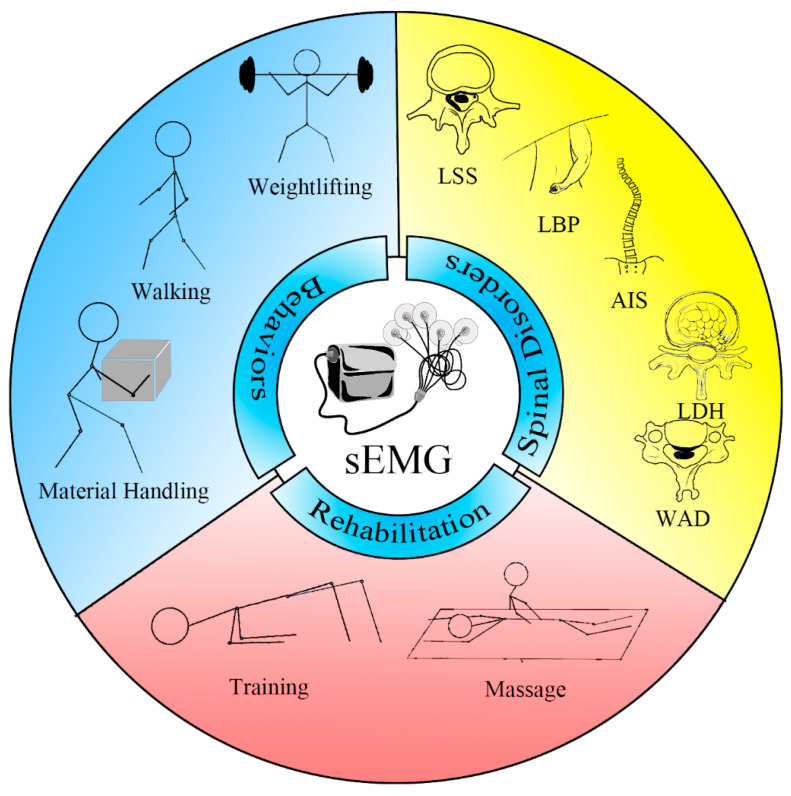
Evaluating paraspinal muscles with sEMG. AIS = adolescent idiopathic scoliosis; LBP = low back pain; LSS = lumbar spinal stenosis; LDH = lumbar disc herniation; sEMG = surface electromyography; WAD = whiplash-associated disorder.

**Table 1 diagnostics-14-01086-t001:** sEMG evaluating the patients with LBP.

Years	First Author	Samples	Conclusions
1983	Cram [4]	66	The differences between both sides’ sEMG activity in the paraspinal muscle groups were important to distinguish LBP patients.
1992	Lee [45]	39	There was a good reproducibility for evaluating patients with LBP by sEMG.
1993	Cassisi [98]	21	Maximum surface integrated electromyography could be helpful for the classification of chronic LBP during isometric exercise.
1993	Cooper [82]	39	The muscle excess fatigue might be peripheral in origin.
1995	Jalovaara [86]	43	sEMG was a valid tool for indirectly assessing pain in LBP patients.
2001	Lu [81]	40	There was a difference in muscle activity patterns between healthy persons and patients with LBP.
2004	Crossman [84]	67	There was non-existent histomorphometric discrepancy between the subjects with and without CLBP.
2005	Humphrey [97]	350	Half-width, initial median frequency, and peak amplitude were great parameters to distinguish the individuals with and without CLBP.
2005	Kuriyama [93]	44	The paraspinal muscles played a role in spinal stabilization.
2015	Adeyemi [99]	47	The activity of paraspinal muscles in the lower back was more influenced by backpack carriage than the upper back among schoolchildren.
2015	Djordjevic [83]	73	There was a significant relationship between the sEMG signal and relative thickness change in the transversal abdominal.
2015	Sánchez-Zuriaga [92]	30	Patients with LBP during pain-free periods displayed changes in EMG activity and reduced maximum lumbar flexion ranges, distinguishing them from non-LBP subjects.
2018	Becker [19]	30	There was a high relationship in lumbar erector spinae between the changes of activity and the degree of function.
2018	Chiou [85]	15	Spectral characteristics of sEMG reflected muscle function.
2018	Yang [87]	56	Less activity of lumbar multifidus muscles observed in the subjects with LBP might contribute to decrease the lumbar stabilization during stoop lift.
2019	da Silva [88]	40	Less lumbar muscle activity and more co-activation between rectus adominis and multifidus muscles in the patients with LBP during one-legged stance task.
2019	Martinez-Valdes [100]	18	Magnitude of activation and the distribution of erector spinae activity were observed as changed in LBP patients.
2019	Sanderson [80]	26	Individuals without symptoms exhibited a spatial redistribution of lumbar erector spinae muscle activity during an endurance task, whereas this adaptation was diminished in the subjects with LBP.
2020	Hao [101]	40	The uneven spatial distribution and asymmetry of lumbar muscle activity were significant factors in CLBP patients.
2020	Shah [90]	23	There was a significant increase in the recruitment of the lumbar multifidus muscle with increased lumbar lordosis in patients with CLBP during quadruped exercise.
2021	Balasch-Bernat [95]	75	Patients with continuous CLBP exhibited elevated EMG activity in the erector spinae and multifidus muscles during the isometric and concentric phases of back extension exercises, as compared to healthy individuals. This difference in muscle activity was also present to a lesser degree when compared to patients with RLBP.
2021	Suehiro [96]	34	In individuals with recurrent LBP, the activation of contralateral erector spinae occurred with a delay compared to those without recurrent LBP during both right and left active hip abduction tests.
2021	Zou [21]	40	Core stability training had the potential to ameliorate symptoms, enhance the fatigue resistance of core muscles, and promote the balance of bilateral multifidus muscle function in patients suffering from nonspecific LBP.
2022	Arvanitidis [94]	30	Patients with CLBP were unable to increase the common fluctuations in torque and high-density sEMG activity during exertion of higher lumbar extension forces.
2022	Varrecchia [91]	23	The individuals with LBP engaged in greater co-activation of their trunk muscles than healthy controls, via the implementation of a fatiguing trunk-stiffening strategy.

Note: sEMG: surface electromyography, LBP: low back pain, CLBP: chronic low back pain, RLBP: recurrent low back pain.

**Table 2 diagnostics-14-01086-t002:** sEMG evaluating the patients with LDH or LSS.

Disorders	Years	First Author	Samples	Conclusions
LDH	2001	Leinonen [110]	35	Chronic pain could impair lumbar feedforward control in CLBP patients.
	2016	Ramos [109]	60	Increased fatigue of the lumbar multifidus compared with the healthy controls.
	2019	Li [111]	30	Percutaneous endoscopic lumbar discectomy for individuals with LDH could normalize paraspinal muscle activation during lumbar flexion–extension movement.
	2020	Zhao [23]	70	There was an imbalance in myoelectric activity in the individuals with chronic LDH, and the muscle strength on the affected side was significantly reduced.
LSS	2003	Leinonen [113]	25	The endurance of paraspinal muscle in LSS patients was good.
	2013	Kääriäinen [115]	60	The activation of paraspinal muscles could be observed in the patients with LSS, which indicated an extensive loss of motor functions in the patients with LSS.
	2016	Kääriäinen [114]	30	During two years of follow-up after decompressive surgery, muscle activation profiles tended to further deteriorate.
	2023	Nüesch [112]	39	During midstance, higher activation of multifidus and erector spinae showed in the patients with LSS compared with the healthy controls.

Note: sEMG: surface electromyography, LDH: lumbar disc herniation, LSS: lumbar spinal stenosis, CLBP: chronic low back pain.

**Table 3 diagnostics-14-01086-t003:** sEMG evaluating the patients with AIS.

Years	First Author	Samples	Conclusions
2002	Lu [118]	34	Unbalanced sEMG activity in the paravertebral muscles could be found in the patients with AIS, which could be decreased after spine fusion.
2004	Perret [129]	16	The presence of short-latency responses and later activities in sEMG after a postural perturbation.
2005	Cheung [121]	23	sEMG of the paraspinal muscles could forecast the progression in idiopathic scoliosis.
2005	Gaudreault [7]	16	There might be a potentially difference at the lower level of the spine
2010	Tsai [124]	74	During isokinetic flexion and extension exercises, the sEMG activities of the thoracic muscle were significantly higher on the concave side than on the convex side in AIS patients with larger curves, which was different from the healthy control group and those with AIS with smaller curves.
2014	Farahpour [128]	20	Asymmetry of muscle activity in AIS patients relied on the direction of the perturbation during postural perturbation.
2018	Ko [125]	25	Asymmetric spinal stabilization exercise could improve the severity of scoliosis, especially at the concave side of paraspinal muscles.
2019	Yuan [116]	90	The ratios of RMS on paraspinal muscles and relaxation time in the scoliotic patients were found greater than non-scoliotic patients.
2021	Park [117]	101	There was a different characteristic of sEMG in different types of adolescent idiopathic scoliosis curves.
2022	Cheung [123]	7	sEMG biofeedback posture training could reduce the asymmetric paraspinal muscle activities and the curve progression.
2022	He [127]	21	The asymmetric and symmetric exercises elicited greater sEMG activity on the convex and concave side, respectively.
2022	Liang [122]	106	The dynamic asymmetric of the erector spinae group of muscles could predict scoliosis aside.
2023	Chan [120]	30	Imbalance in paraspinal muscles could play a potential role in the rehabilitation for AIS patients.

Note: sEMG: surface electromyography, AIS: adolescent scoliosis.

**Table 4 diagnostics-14-01086-t004:** sEMG evaluating the patients with SCI or NP or WAD.

Disorders	Years	First Author	Samples	Conclusions
SCI	2020	Singh [130]	36	Children with SCI exhibited compromised trunk control, which affected their ability to activate trunk muscles both above and below the level of injury.
	2022	Hoglund [131]	15	Different stimulations resulted in different responses of different paraspinal muscles in the patients with SCI.
NP	1994	Nanno [135]	96	The cervical intermittent traction could contribute to relieve pain, increase the frequency of electromyographic signals, and improve blood flow in affected muscles.
	2008	Lecompte [133]	27	There was a difference in the muscles’ function between the fighter pilots with and without NP.
	2017	Park [134]	40	Significantly altered neck motion and muscle activation during active neck extension were observed in the patients with unilateral posterior NP.
	2021	Sremakaew [132]	50	Except for the upper trapezius, there was higher activity in all muscles in NP patients.
WAD	2003	Siegmund [136]	44	Habituation might be a potential confounder of whiplash injury studies using repeated perturbations.
	2007	Descarreaux [138]	31	The patients with WAD were able to produce isometric forces with spatial precision in which the time to peak force was increased.

Note: sEMG: surface electromyography, SCI: spinal cord injury, NP: neck pain, WAD: whiplash injury.

## Data Availability

Data sharing is not applicable to this article as no new data were created or analyzed in this study.

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
