# Peer review of "The Application of Surface Electromyography Technology in Evaluating Paraspinal Muscle Function"

_diagnostics, 2024, doi:10.3390/diagnostics14111086_

Round 1

Reviewer 1 Report

Comments and Suggestions for Authors

Dear, author

   I appreciated your review about this topic.  This is the good review in this topic.  But I suggest you to mention about limitation in using this sEMG in clinical practice.

Comments on the Quality of English Language

Quality of English is very good

Reviewer 2 Report

Comments and Suggestions for Authors

Authors presented a review on the EMG technology finalized to the paraspinal muscles. The review proposed by authors is interesting and relevant for the journal. However, I think a better description on how authors conducted the search should be added.

Following are some comments that I think may improve the quality of the review.

The characteristics that you used for the search are not defined:

On which websites did authors looked for relevant papers (Pubmed, Web of Science, Scopus)?

What was the query that they used?

How many papers did they found and what were the inclusion/exclusion criteria?

During which day was the search performed, to be sure that no more papers could be found before that day?

Line 52: Another important application of EMG, that I think it may be worth of mention in the introduction, is the control of robotic devices, prostheses, or exoskeletons (for your literature search you may start from: Singh et al., 2012, Trends and challenges in EMG based control scheme of exoskeleton robots-a review; Borzelli et al., 2020, Identification of the best strategy to command variable stiffness using electromyographic signals).

Line 647: some studies, which should be mentioned, already tried to define the best practiced for EMG elaboration or the best electrodes placement of some muscles, i.e. the SENIAM project.

Please add examples of papers that in line with the future directions that you describe in the conclusion. E.g. Line 651: Clinical application of EMG.

Line 655: Machine learning and EMG

Line 663: EMG and biofeedback
